# QUERY AND RESPONSE AUGMENTATION CANNOT HELP OUT-OF-DOMAIN MATH REASONING GENERALIZATION

## ABSTRACT

In math reasoning with large language models (LLMs), fine-tuning data augmentation by query evolution and diverse reasoning paths is empirically verified effective, profoundly narrowing the gap between open-sourced LLMs and cutting-edge proprietary LLMs. In this paper, we conduct an investigation for such data augmentation in math reasoning and are intended to answer: (1) What strategies of data augmentation are more effective; (2) What is the scaling relationship between the amount of augmented data and model performance; and (3) Can data augmentation incentivize generalization to out-of-domain mathematical reasoning tasks? To this end, we create a new dataset, AugGSM8K, by complicating and diversifying the queries from GSM8K and sampling multiple reasoning paths. We obtained a series of LLMs called MuggleMath by fine-tuning on subsets of AugGSM8K. MuggleMath substantially achieves new state-of-the-art on GSM8K (from 54% to 68.4% at the scale of 7B, and from 63.9% to 74.0% at the scale of 13B). A log-linear relationship is presented between MuggleMath's performance and the amount of augmented data. We also find that MuggleMath is weak in out-of-domain math reasoning generalization to MATH. This is attributed to the differences in query distribution between AugGSM8K and MATH which suggest that augmentation on a single benchmark could not help with overall math reasoning performance.

## 1 INTRODUCTION

The emergence of large language models (LLMs) (Ouyang et al., 2022; Anil et al., 2023; OpenAI, 2023) has profoundly revolutionized the field of natural language processing, exhibiting versatile performance in various tasks like code generation (Chen et al., 2021; Luo et al., 2023b), instruction following (Longpre et al., 2023), long context understanding (Tworkowski et al., 2023), and math reasoning (Wei et al., 2022; Taylor et al., 2022; Lewkowycz et al., 2022a). Math reasoning as a representative reasoning task is widely studied to access the reasoning abilities in LLMs (Cobbe et al., 2021; Hendrycks et al., 2021). Proprietary LLMs, such as GPT-3.5, and GPT4 (OpenAI, 2023) have shown exceptional mathematical reasoning abilities, while there remains a substantial gap between open-source LLMs, such as GPT-J (Wang & Komatsuzaki, 2021) and LLaMA (Touvron et al., 2023a;b)) and the cutting-edge proprietary models.

To enable better mathematical reasoning abilities in open-sourced LLMs, they generally undergo a fine-tuning stage on supervised reasoning datasets. A series of efforts are committed to enhancing the mathematical reasoning capabilities of open-source LLMs, where a mainstream approach involves first augmenting new mathematical problems and answers, followed by supervised fine-tuning on the augmented dataset (Yuan et al., 2023a; Luo et al., 2023a; Yu et al., 2023). This type of approach has achieved good results, and in this paper, we would like to explore what are the key factors affecting the effectiveness of data augmentation for mathematical reasoning tasks and the scaling relationship between the amount of data augmentation and model performance. Specifically, with the help of proprietary models (GPT-3.5 and GPT-4), we applied five types of mathematical problem augmentation methods based on human experience in creating variations of mathematical problems similar to Luo et al. (2023b;a). We further generated multiple reasoning paths for each augmented problem since distinct reasoning paths can also enhance chain-of-thought reasoning (Huang et al., 2022; Zhu et al., 2023a; Yuan et al., 2023a) (as the left sub-figure of Fig. 1). We obtained a new dataset called AugGSM8K after data augmentation on a widely used mathemati-

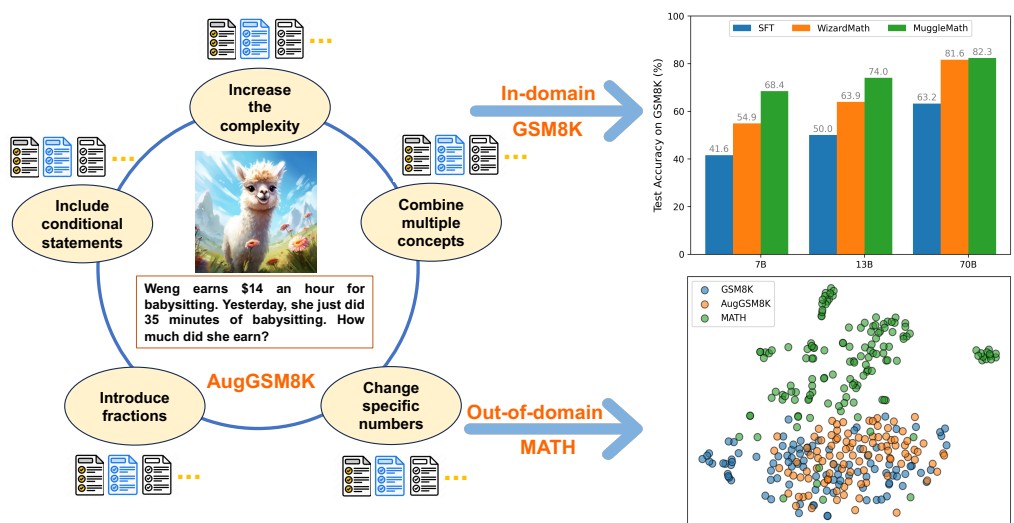

Figure 1: Overview of AugGSM8K, which is obtained by applying different methods of data augmentation to the queries in the GSM8K training set and sampling different reasoning paths for each augmented query, the performance compare of SFT, WizardMath, and MuggleMath on in-domain dataset GSM8K and the visualized query distribution of GSM8K, AugGSM8K, and MATH. MuggleMath is a series of models derived from fine-tuning LLaMA-7B, LLaMA-2-7B, LLaMA-2-13B, and LLaMA-2-70B on subsets of AugGSM8K.

cal reasoning dataset GSM8K (Cobbe et al., 2021). By supervised fine-tuning on the open-source LLaMA (Touvron et al., 2023a) and LLaMA-2 (Touvron et al., 2023b) LLMs on different subsets of AugGSM8K, we obtained a series of models dubbed MuggleMath. We find that with sufficient amounts of data, MuggleMath achieves a new state-of-the-art on GSM8K. In addition to this, we find a log-linear relationship between the performance of MuggleMath and the amount of data augmentation over a range of data volumes. Although MuggleMath achieves strong performance on the GSM8K test set, the rationales for performance improvement by data augmentation remain unclear. We are therefore interested in the specific reason behind the performance improvement and whether it brings enhancement in LLMs' mathematical reasoning capabilities generally.

To validate the generalization of MuggleMath, we conduct multi-task learning and analyze the transferability with AugGSM8K and MATH. We found that LLMs trained with supervised learning after data augmentation on GSM8K only bring marginal improvements to performance on MATH (Hendrycks et al., 2021). By visualizing the data distribution in the embedding space of LLaMA-2-7B (as the bottom right sub-figure of Fig. 1), we observe that the embedding distribution of problems in AugGSM8K is very close to that of GSM8K, but significantly different from the problem distribution in the MATH dataset. This is the fundamental reason why performance improvements from data augmentation on GSM8K cannot be generalized to MATH.

The main contributions of our work can be summarized as follows:

- By augmenting GSM8K with various queries and multiple reasoning paths, we curated GSM8K to a new dataset named AugGSM8K.
- We utilize AugGSM8K for fine-tuning the LLaMA models to obtain MuggleMath, which greatly improves the in-domain performance of the open-sourced LLMs on GSM8K and achieves new state-of-the-art performances.
- We find a log-linear relationship between the accuracy of the model on the test set and the amount of data augmentation within a certain range while the coefficient is similar to augmenting new human-written samples.
- We demonstrate that the performance gains from data augmentation on GSM8K are difficult to generalize to out-of-domain dataset MATH due to distribution differences.

## 2 RELATED WORKS

**Mathematical Reasoning for Large Language Models.** Mathematical reasoning is a crucial ability to examine large language models (Cobbe et al., 2021; Hendrycks et al., 2021; Wei et al., 2022; Yuan et al., 2023b). The mathematical reasoning ability of LLMs can be enhanced by math-related pre-training (Hendrycks et al., 2021; Lewkowycz et al., 2022a; Taylor et al., 2022; Lightman et al., 2023) and math-related supervised fine-tuning (Yuan et al., 2023a; Luo et al., 2023a; Yue et al., 2023; Yu et al., 2023). Query augmentation (Luo et al., 2023a; Yu et al., 2023) and response augmentation (Huang et al., 2022; Zelikman et al., 2022; Ni et al., 2023; Zhu et al., 2023b; Yuan et al., 2023a) are useful techniques to improve math in-domain performances during SFT. Query augmentation methods usually generate rephrased, easier, or harder problems and use proprietary LLMs to generate answers. Response augmentation methods generate new reasoning paths for problems in the training set. They could rely on answers in the training set to filter the generated reasoning paths. Yuan et al. (2023a) invests the scaling relationship on supervised LLMs math performance with pre-train loss, supervised data amount, and augmented reasoning path amount. Yu et al. (2023) is a contemporary work that is very similar to us in the augmentation method. Compared with these two works, our work investigates the quantitative relationship between query and response augment amounts and in-domain and out-of-domain performances.

**Data Augmentation for LLM.** Data augmentation is a common technique to improve downstream task performance in NLP (Feng et al., 2021). In the era of large language models, data augmentation is usually used for generating instruction following SFT datasets (Wang et al., 2023b; Taori et al., 2023). Queries (Ding et al., 2023; Xu et al., 2023) and responses (Mukherjee et al., 2023) of SFT datasets can both be augmented by prompting state-of-the-art proprietary LLMs. Compared with their work, we are concentrated on augmenting math SFT dataset and we are more interested in scaling relationships on in-domain and out-of-domain generalizations.

**Out-of-Distribution Generalization.** The challenge of out-of-distribution (OOD) generalization has garnered widespread attention across various domains(Karras et al., 2018; Wang et al., 2021; Zhou et al., 2023) in machine learning. This issue arises when the distribution of data encountered by a model during testing diverges from that of the training phase, leading to a decline in model performance. The OOD problem is multifaceted(Lipton et al., 2018; Schölkopf et al., 2012; Tran et al., 2022; Cai et al., 2023), with subcategories such as covariate shifts and concept shifts, among others. To mitigate the effects of OOD scenarios, a diverse array of strategies has been developed, including unsupervised domain generalization(Wang et al., 2021; Zhou et al., 2023), stable learning(Shen et al., 2020; Kuang et al., 2020), invariant representation learning(Creager et al., 2021), causal learning(Peters et al., 2015), and invariant risk minimization(Mao et al., 2023) and etc. Recent trends in the community have shown a growing preference for performance enhancement through data augmentation during the Self-supervised Fine-tuning (SFT) stage in large-scale models. However, the extent of OOD issues associated with this method and their severity remain underexplored. This study aims to fill this gap by conducting empirical experiments and providing a visual analysis to assess the impact of data augmentation on OOD generalization in the context of large models.

## 3 EXPERIMENTS

We first introduce our experimental setup (§3.1) and dataset augmentation method (§3.2).

### 3.1 EXPERIMENTAL SETUP

**Problem Definition.** We define the math reasoning SFT dataset as $\mathcal{D} = \{q_i, a_i\}_i$, where $q_i$ is a question and $a_i$ is a reasoning path with an answer. We augment the SFT dataset to a new dataset $\mathcal{D}' = \{q'_i, a'_i\}_i$. We apply SFT on the pre-trained language models and measure the augmented SFT dataset based on the accuracy of the in-domain test set $\mathcal{D}_{in}$ and out-of-domain test set $\mathcal{D}_{out}$. We calculate the accuracy based on greedy decoding.

**In-domain Dataset.** GSM8K (Cobbe et al., 2021) is a dataset with elementary school math word problems with 7,473 training problems and 1,319 testing problems. This is viewed as our in-domain dataset $\mathcal{D}$ and we will augment and train on it. The test set of GSM8K is viewed as $\mathcal{D}_{in}$.

**Out-of-domain Dataset.** MATH (Hendrycks et al., 2021) is a dataset with challenging high-school math problems. Problems are classified into the following topics: Prealgebra, Algebra, Number Theory, Counting and Probability, Geometry, Intermediate Algebra, and Precalculus. Problems in MATH are harder and more diverse than in GSM8K. We use 500 test problems from Lightman et al. (2023) as our out-of-domain test dataset $\mathcal{D}_{out}$. The reason we do not use MATH as the in-domain dataset is that it is hard to apply query and response augmentation for MATH-level problems since GPT-4 can only have an accuracy of 42.

**Training.** We employ state-of-the-art open-source LLMs for fine-tuning, including LLaMA-1 7B (Touvron et al., 2023a), LLaMA-2 7B, LLaMA-2 13B, and LLaMA-2 70B (Touvron et al., 2023b), all of which undergo full fine-tuning. We adopt system prompt from Taori et al. (2023) for fine-tuning and listed in Appx. §A. We use AdamW for optimization. The training proceeds for three epochs with a learning rate of 2e-5, a warmup ratio of 0.03, and a cosine learning rate scheduler. We do not apply early stops to choose checkpoints. The computational hardware includes 8 NVIDIA A100 GPUs for 7B and 13B models and includes 32 NVIDIA A100 GPUs for 70B models.

### 3.2 DATASET AUGMENTATION

**Query Augmentation.** We use `gpt-3.5-turbo-0613` and `gpt-4-0613` to generate new queries. Inspired by Evol-Instruct (Luo et al., 2023b;a), we find that the diversity and complexity of queries in augmented datasets play a vital role in improving math reasoning benchmark performance. We employ human expertise and knowledge in modifying mathematical problems for query augmentation. Below are five query augmentation methods used in our experiments: **Change specific numbers; Introduce fractions or percentages; Combine multiple concepts; Include a conditional statement; Increase the complexity of the problem.** The examples and detailed prompts we used for query augmentation are listed in Appx. §B. The examples of augmented queries are shown in Tab. 12.

**Response Augmentation.** We use `gpt-3.5-turbo-0613` and `gpt-4-0613` to augment more reasoning paths instead of using the trained SFT model proposed in Yuan et al. (2023a). The main reason is we can not filter out wrong reasoning paths without final answers. Thus we need to use a model that is as accurate as possible which is the state-of-the-art LLMs *ChatGPT*. We use a 1-shot prompt to ensure augmented response formats. The response prompt we used for query augmentation is listed in Appx. §C. Augmented responses can result in some unconventional answers, such as excessively long reasoning paths and reasoning paths that do not contain an answer at their end. We devise manual rules to filter out these corresponding query-response pairs and manual rules are detailed in Appx. §D. The examples of augmented responses are shown in Tab. 13.

**Augmented Dataset.** The original GSM8K training set has 7,473 samples. We augment 5 more queries for each query in the training set and yield $7,473 \times 5 = 37,365$ augmented queries. We run this query augmentation three times with $\mathcal{D}_1, \mathcal{D}_3$ by GPT-3.5 and $\mathcal{D}_2$ by GPT-4, and $\|\mathcal{D}_i\| = 37,365$. Then we generate one response for each augmented query for $\mathcal{D}_i$ and apply response filtering. We consider the query-response pairs after filtering as $\mathcal{D}_i^j$. We obtain approximately 30,000 query-response pairs for each $\mathcal{D}_i^j$. To explore the performance differences of different augmented settings, we generate five responses on the augmented queries $\mathcal{D}_1$ with GPT-4's temperature set to 1.0 ($\mathcal{D}_1^1 \sim \mathcal{D}_1^5$), one response with GPT-4's temperature set to 0.0 ($\mathcal{D}_1^6$), and one response with GPT-3.5's temperature set to 1.0 ($\mathcal{D}_1^7$). We also try a zero-shot response generation named $\mathcal{D}_1^8$. We use GPT-4 to augment responses as $\mathcal{D}_2^1, \mathcal{D}_3^1$. Since $\mathcal{D}_2^1$ is significantly larger than other subsets, we downsample it to $\hat{\mathcal{D}}_2^1$. We refer to the union of all augmented data and the original GSM8K training set as AugGSM8K, upon which we conduct experiments using various subsets. Detailed augmented dataset notations are listed in Tab. 1.

## 4 FINDING AND DISCUSSION

In this section, we conduct analyses spanning several aspects of data augmentation for mathematical reasoning, including in-domain generalization (§4.1), out-of-domain generalization (§4.2), the relationship between training (§4.3) and testing performance, and augmentation on hard problems (§4.4). We present the main experimental findings and provide delve-deep discussion accordingly.

| Subset | Query | Response | Temp. | Size (K) |
|---|---|---|---|---|
| $\mathcal{D}$ | - | - | - | 7.5 |
| $\mathcal{D}_1^1 \sim \mathcal{D}_1^5$ | GPT-3.5 | GPT-4 | 1 | 30 |
| $\mathcal{D}_1^6$ | GPT-3.5 | GPT-4 | 0 | 30 |
| $\mathcal{D}_1^7$ | GPT-3.5 | GPT-3.5 | 1 | 25 |
| $\mathcal{D}_1^8$ | GPT-3.5 | GPT-4 | 1 (zero-shot) | 30 |
| $\mathcal{D}_2^1$ | GPT-4 | GPT-4 | 0 | 35 |
| $\hat{\mathcal{D}}_2^1$ | GPT-4 | GPT-4 | 0 | 30 |
| $\mathcal{D}_3^1$ | GPT-3.5 | GPT-4 | 1 | 30 |

Table 1: The description of different subsets of the augmented in-domain dataset AugGSM8K.

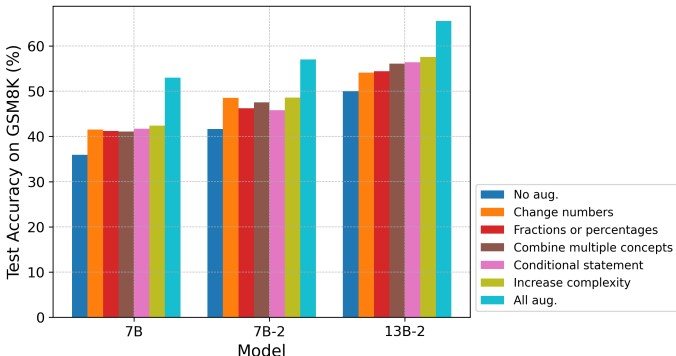

Figure 2: Comparison of test set accuracy on GSM8K for models of varying scales after fine-tuning on AugGSM8K subsets with different query augmentation strategies.

## 4.1 IN-DOMAIN SCALING ON DIFFERENT AUGMENTATION

**Query Augmentation Types.** We want to examine whether query augmentation works for math reasoning SFT since Luo et al. (2023a) applies PPO which cannot make an apple-to-apple comparison. Each query in the original training dataset is augmented with 5 different types. We cluster these queries based on the query types. We apply SFT on the original training set ($\mathcal{D}$), each query type augmentation, and a combination of them ($\mathcal{D} + \mathcal{D}_1^1$). Results are shown in Fig. 2 and Tab. 6. Compared with no augment, each query augment method can improve the in-domain performance. **Increase complexity** augmentation method improves most among all of them. This suggests that enhancing the complexity of queries is one of the key factors influencing the sample efficiency of data augmentation. Combining five augment methods yields the best in-domain performance which proves the effectiveness of Evol-instruct under a fair setting.

**Query Augmentation Amount.** We examine how query augmentation amount affects the in-domain performance. We examine seven data volume configurations including partitioning $\mathcal{D}_1^1$ into proportions of 0, 0.2, 0.4, 0.6, 0.8, 1.0, as well as $\mathcal{D}_1^1 + \hat{\mathcal{D}}_2^1$ and $\mathcal{D}_1^1 + \hat{\mathcal{D}}_2^1 + \mathcal{D}_3^1$ as the augmented datasets. Each augmented query only has one augmented response. They are mixed with GSM8K $\mathcal{D}$ to apply SFT. From Fig. 3 and Tab. 7, we can find that within the data volume range of 13-97K, the in-domain performance exhibits a log-linear relationship with the query amount. We employ linear regression to approximate this relationship. As shown in Tab. 2, pre-training models with better initial math reasoning capabilities exhibit a smaller slope which is consistent with Yuan et al. (2023a). This suggests it is harder to improve reasoning ability for a better pre-trained model. We also conduct validations on our fitted scaling law with an interpolate point at a query amount of 17K ($\mathcal{D}+\mathcal{D}_1^1 \times 0.3$) and an extrapolate point at a query amount of 104K ($\mathcal{D}+\mathcal{D}_1^1+\hat{\mathcal{D}}_2^1+\mathcal{D}_3^1$), discovering that the regression offers accurate predictions of model performances. We should notice this scaling law cannot be correct within all dataset size ranges since the test set accuracy is bounded.

Besides, the fitted regression shows when **query augmentation amount** doubles, LLaMA-7B models will improve $10.7 \times \log(2) = 7.4$, LLaMA2-7B will improve $9.8 \times \log(2) = 6.8$, and LLaMA2-

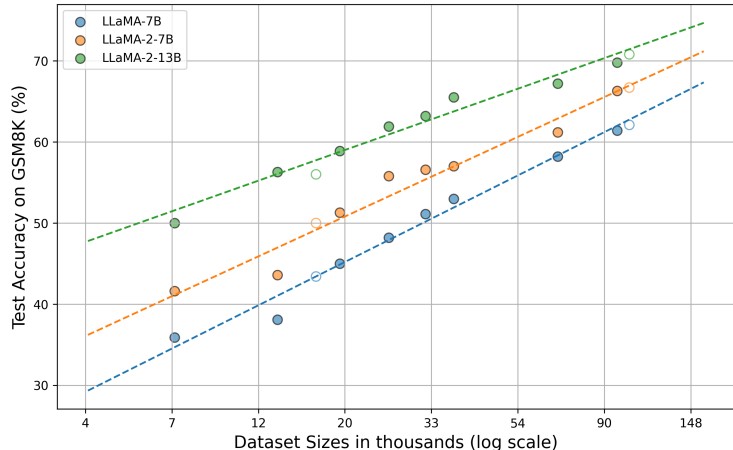

Figure 3: Comparison of test set accuracy on GSM8K for models of varying scales after fine-tuning on AugGSM8K subsets with different query augmentation amount.

13B will improve $7.6 \times \log(2) = 5.3$. As shown in Yuan et al. (2023a), it is estimated that when **human-written sample amount** doubles, LLaMA-7B models will improve 6.5 score, LLaMA2-7B will improve 6.6 score, and LLaMA2-13B models will improve 5.5 score. Query augmentation is similarly effective to human-written samples in term of in-domain performance. This demonstrates that query augmentation benefits from the performing proprietary LLMs on GSM8K, thus the sample quality generated by query augmentation is as high as those of human-written samples.

| Model | 7B | 7B-2 | 13B-2 |
|---|---|---|---|
| Estimation | $y = 10.7 \log(x) + 13.2$ | $y = 9.8 \log(x) + 21.3$ | $y = 7.6 \log(x) + 36.3$ |
| x = 17 prediction | 43.4 | 49.2 | 57.7 |
| x = 17 observation | 43.4 | 50.0 | 56.0 |
| x = 104 prediction | 62.7 | 67.0 | 71.4 |
| x = 104 observation | 62.1 | 66.7 | 70.8 |

Table 2: The scaling law on amounts of augmented query in GSM8K.

**Response Augmentation Amount.** We further investigate under the data augmentation setting, if we keep the number of queries constant and increase the number of responses, how the in-domain performance changes. We use $\mathcal{D}_1$ as augmented queries and vary the response amount from 1 to 5 per augmented query. We also try majority voting (Wang et al., 2023a; Huang et al., 2022) to filter the augmented response since we cannot know the correct answer. In Fig. 4 and Tab. 8, we find for LLaMA-7B and LLaMA-2-7B models, once the response data volume reaches 97K (3 responses per query), further increase the number of responses do not yield performance improvement. Before this point, model performance improves as the response amount increases. Thus, query augmentation with accurate responses seems more effective than only response augmentation with the augmented data size scales up. As for the LLaMA-2-13B model, within the data volume range of 37K to 157K, model performance consistently rises in a roughly linear fashion with increasing data volume, with a slower rate than the 7B model within the ascending interval. We then investigate the performance impact brought by majority voting filtering. If all responses have different answers, we discard the corresponding query-response. Surprisingly, we find that after applying majority voting, the model performance at the same data scale is generally lower than not applying filtering. A possible explanation is that even wrong responses generated by GPT-4 are useful for worse models (LLaMA) to improve their abilities. Another explanation is the reduction in the number of queries, as we discard the corresponding query when all response answers are different.

**Query and Response Sources.** Here we examine how the query and response quality influence the augmented model performance. We list results in Tab. 3, and draw the following conclusions: (a)

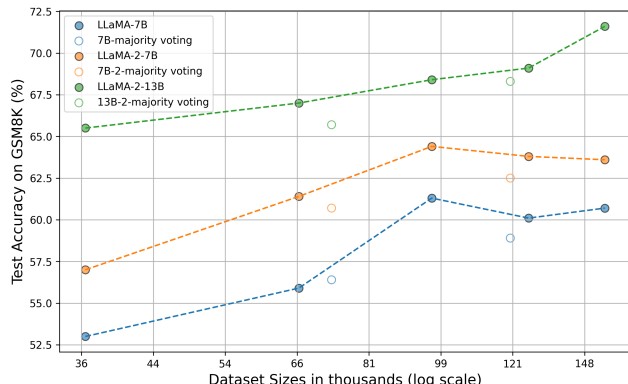

Figure 4: Comparison of test set accuracy on GSM8K for models of varying scales after fine-tuning on AugGSM8K subsets with different response augmentation amount.

| Model | 7B | 7B-2 | 13B-2 |
|---|---|---|---|
| $\mathcal{D}$ | 35.9 | 41.6 | 50.0 |
| $+\mathcal{D}_1^1 \times 0.8$ | 51.1 | 56.6 | 63.2 |
| $+\mathcal{D}_1^1$ | 53.0 | 57.0 | 65.5 |
| $+\mathcal{D}_1^6$ | 51.6 | 58.0 | 63.8 |
| $+\mathcal{D}_1^7$ | 41.3 | 46.7 | 52.8 |
| $+\mathcal{D}_1^8$ | 49.4 | 53.3 | 62.2 |
| $+\hat{\mathcal{D}}_2^1$ | 52.3 | 57.8 | 63.3 |

Table 3: Performance of subsets of AugGSM8K with different query and response sources. $+\mathcal{D}_1^1$ is an omission of $\mathcal{D} + \mathcal{D}_1^1$, and the same notation is used in other tables in this paper.

Comparing $\hat{\mathcal{D}}_2^1$ and $\mathcal{D}_1^6$, we find that the queries generated by GPT-4 and GPT-3.5 have no significant impact on SFT performance. (b) Comparing $\mathcal{D}_1^1$ and $\mathcal{D}_1^6$, we can conclude that when using GPT-4 to generate responses, the temperature has no significant impact on SFT performance. (c) Comparing $\mathcal{D}_1^1$ and $\mathcal{D}_1^8$, we can conclude that, compared to the zero-shots generation method, the response augmentation prompt we propose plays a substantial role in enhancing the quality of the generated data (+3.6 for LLaMA-7B, +3.7 for LLaMA-2-7B, +3.3 for LLaMA-2-13B). The main reason we consider this is our 1-shot setting stabilizes the response format. (d) Comparing $\mathcal{D}_1^7$ (25K) and $\mathcal{D}_1^1 \times 0.8$ (24K), we can conclude that, compared to GPT-3.5, the response augmented using GPT-4 yields significantly better results for SFT.

**Combination.** We investigate how the combination of query augmentation and response augmentation will affect the model's performance. Results are listed in Tab. 4. We conduct SFT on $\mathcal{D} + \sum_{i=1}^3 \mathcal{D}_1^i + \hat{\mathcal{D}}_2^1 + \mathcal{D}_3^1$ and named MuggleMath effectively improving the in-domain accuracy of each model compared to using query or response augmentation only. These models outperform previous state-of-the-art open-sourced models with a very large margin for 7B and 13B models. More comparisions are in appx. §H. Case studies of MuggleMath are listed in Tab. 14. It demonstrates that query augmentation and response augmentation can complement each other to a certain extent to improve in-domain performance.

## 4.2 OUT-OF-DOMAIN GENERALIZATION ON DIFFERENT AUGMENTATION

We have found that query and response augmentation significantly improves in-domain math reasoning performance. But we really interested in whether we can improve performances on out-of-domain distribution. We employ multi-task learning and transfer learning to see how models performed on the MATH dataset. We list results in Tab. 5. We find that (1) Multi-task learning and transfer learning outperform single-task supervised fine-tuning on LLaMA2-7/13B and do not improve on LLaMA-7B. (2) Although augmenting more query and response can improve GSM8K significantly, it has little to no help in improving MATH performance which indicates that in-domain augmentation is **not** helpful for out-of-domain generalization in this setting. Case studies of models

| Model | 7B | 7B-2 | 13B-2 | 70B-2 |
|---|---|---|---|---|
| $\mathcal{D}$ | 35.9 | 41.6 | 50.0 | 63.2 |
| RFT (Yuan et al., 2023a) | 49.1 | 51.2 | 55.3 | 64.8 |
| WizardMath (Luo et al., 2023a) | - | 54.9 | 63.9 | 81.6 |
| MetaMath(Yu et al., 2023) | - | 66.5 | 72.3 | 82.3 |
| $+\mathcal{D}_1^1 + \mathcal{D}_1^2 + \mathcal{D}_1^3$ | 61.3 | 64.4 | 68.4 | - |
| $+\mathcal{D}_1^1 + \hat{\mathcal{D}}_2^1 + \mathcal{D}_3^1$ | 61.4 | 66.3 | 69.8 | - |
| $+\sum_{i=1}^3 \mathcal{D}_1^i + \hat{\mathcal{D}}_2^1 + \mathcal{D}_3^1$ | 65.2 | 67.4 | 72.6 | 80.1 |
| $+\sum_{i=1}^3 \mathcal{D}_1^i + \hat{\mathcal{D}}_2^1 + \mathcal{D}_3^1$ (n=512) | **65.4** (+16.3) | **68.4** (+13.5) | **74.0** (+10.1) | **82.3** (+0.7) |

Table 4: In-domain performance of combination on query and response augmentation. n=512 means that max decode token count is 512. For other experiments, we use max decode token count 256. Experiments on the perturbed test set are list in Appx. §H
.

| Training Setting | 7B | 7B-2 | 13B-2 |
|---|---|---|---|
| *In Context Learning* on MATH | 2.9 | 2.5 | 3.9 |
| *Supervised Fine-tuning* on MATH | 4.8 | 5.8 | 6.0 |
| *Multi-task learning* | | | |
| +MATH | 4.6 | 6.2 | 7.6 |
| $+\mathcal{D}_1^1$+MATH | 4.8 | 4.8 | 8.4 |
| *Transfer learning* | | | |
| $\mathcal{D} \rightarrow$MATH | 4.4 | 6.0 | 9.4 |
| $+\mathcal{D}_1^1 \rightarrow$MATH | 6.2 | 5.6 | 7.8 |
| $+\sum_{i=1}^3 \mathcal{D}_1^i + \hat{\mathcal{D}}_2^1 + \mathcal{D}_3^1 \rightarrow$MATH | 5.6 | 8.4 | 9.4 |
| $+\sum_{i=1}^7 \mathcal{D}_1^i$-majority voting$+\mathcal{D}_2^1 + \mathcal{D}_3^1 \rightarrow$MATH | 5.6 | 6.0 | 9.0 |

Table 5: Comparison of test set accuracy on MATH. Multi-task learning means that we fine-tune the models on the mixed dataset of AugGSM8K subset and MATH. Transfer learning means that we first fine-tune the models on subsets of AugGSM8K and then fine-tune on MATH.

performed on MATH are listed in Tab. 15. To further investigate why AugGSM8K helps little on the MATH dataset, we visualize the hidden representation of problems encoded by LLaMA2-7B using t-SNE in Fig. 5. We find GSM8K and MATH are separated in hidden space and only some of the problems in MATH are laid in the span of GSM8K. The augmented GSM8K problems are laid in the same span of GSM8K which makes sense why it improves little for MATH. This suggests if we want to improve math reasoning benchmark performances for LLMs, we can choose (1) apply augmentation on diverse math subjects since augment on one benchmark may not improve others (2) improve pre-training since we find larger models yield overall better performances.

## 4.3 TRAINING SET VS. TEST SET ACCURACY

Query and response augmentation generate similar problems for the training set which leads to better training set accuracy. We have shown augmentations improve the accuracy of the in-domain test set. We want to investigate the relationship between the accuracy of the training set and the test set to find if the accuracy of the training set can be a performance indicator. We sample 500 samples from the original training set to calculate the accuracy. From Fig. 9, the training and test accuracy generally exhibit a positive correlation across different augmented data which shows the training accuracy could be an indicator of in-domain performance unless deliberately overfitting.

## 4.4 MAKE MORE AUGMENTATION ON HARDER PROBLEMS

During the query augmentation process, it is crucial to understand which kind of queries should be augmented. Augmenting too many easy problems may not be effective since the model may have mastered this level of problems. Here we examine if the model improves more when we augment more on harder or wrong problems. We define **hard** problems based on the number of

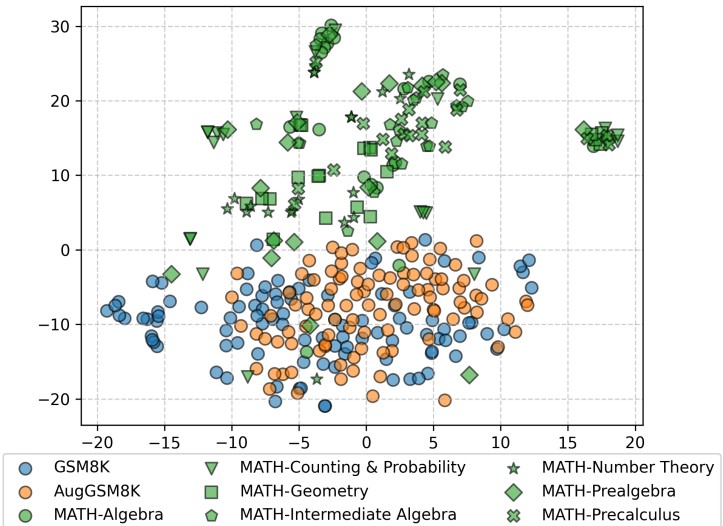

Figure 5: The embedding visualization of queries in GSM8K, MATH and AugGSM8K.

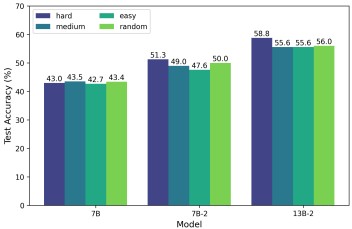

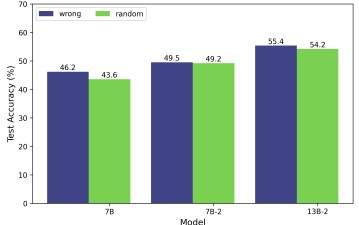

Figure 6: The performance of SFT models with different difficulty augmentation on GSM8K.

Figure 7: The performance of SFT models with wrong problem augmentation on GSM8K.

equations, specifically, problems with fewer than three reasoning steps as easy, those with exactly three steps as medium, and those with more than three steps as hard(see more details in Appx. §G). We define **wrong** problems if the SFT model solves them incorrectly. We apply SFT on subsets of AugGSM8K with augmented queries on easy, medium, hard, wrong, and random problems. From Fig. 6 and Tab. 10, it is evident that for LLaMA-2-7B and LLaMA-2-13B, the performance gain from augmenting hard problems is significantly higher than that from augmenting other types of problems. From Fig. 7 and Tab. 11, we find that augmenting incorrect problems on three models consistently improves more than random query augmentation.

## 5 CONCLUSION

In this paper, we investigate the scaling property of query and response augmentation with respect to math reasoning in-domain and out-of-domain performance. We find that query and response augmentation can improve in-domain performance very effectively which has a similar improvement to human-written query-response pairs augmentation. Although we can obtain state-of-the-art in-domain performance by such augmentation, we find it cannot generalize to out-of-domain math reasoning performances. Since we cannot enumerate all math-related benchmarks and augment all of them, improving pre-training seems a practicable approach to improve math reasoning for LLMs.

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

## A INSTRUCTION PROMPT FOR TRAINING AND INFERENCE

Here is the instruction prompt used for the training and inference stage.

> **Fine-tuning system prompt**
>
> Below is an instruction that describes a task. Write a response that appropriately completes the request.### Instruction: **Query.** ### Response:

## B QUERY AUGMENTATION PROMPT

Here is the query augmentation prompt we used. We require the models to generate five different augmented problems with our provided example. We use `gpt-3.5-turbo-0613` and `gpt-4-0613` APIs with a temperature of 1.0 to obtain augmented problems.

> **Query augmentation prompt**
>
> I want you to act as a math teacher. I will provide a grade school math question and you will help to to create more challenging math questions by given ways. Given the question: "James writes a 3-page letter to 2 different friends twice a week. How many pages does he write a year?", you will modify it by following ideas:
> 1. **Change specific numbers**: James writes a 2-page letter to 2 different friends 3 times a week. How many pages does he write in 4 years?
> 2. **Introduce fractions or percentages**: James writes a 3-page letter to 2 different friends twice a week. Each week, he adds 50% more pages to each letter. How many pages does he write in a month?
> 3. **Combine multiple concepts**: James writes a 3-page letter to 2 different friends twice a week. He uses both sides of the paper and each side can hold 250 words. If James writes 100 words per minute, how long does it take for him to write all the letters in a week?
> 4. **Include a conditional statement**: James writes a 3-page letter to 2 different friends twice a week. If it's a holiday, he writes an additional 5-page letter to each friend. Considering there are 10 holidays in a year, how many pages does he write in a year?
> 5. **Increase the complexity of the problem**: James writes a 3-page letter to 2 different friends twice a week. In addition, he writes a 5-page letter to 3 other friends once a week. How many pages does he write in a month, assuming there are 4 weeks in a month?
> Now you are given the question:
> **A new math problem here.**

## C RESPONSE AUGMENTATION PROMPT

We use this prompt to generate responses to ensure the response format which can be viewed as 1-shot setting. We use `gpt-3.5-turbo-0613` and `gpt-4-0613` with temperature 0.0 or 1.0.

> **Response augmentation prompt**
>
> I want you to act as an excellent math solver. You will solve the given math question step by step. You need to reply with a python dictionary in the same format as the given examples. Retain decimals to three decimal places. The formulas in the process need to use the format: 48/2 = <<48/2=24>>24 clips. The end of response needs to be: #### {answer}.
> Examples: {"query": "Natalia sold clips to 48 of her friends in April, and then she sold half as many clips in May. How many clips did Natalia sell altogether in April and May?", "response": "Natalia sold 48/2 = <<48/2=24>>24 clips in May.Natalia sold 48+24 = <<48+24=72>>72 clips altogether in April and May.#### 72"}.
> The given question:
> **A new math problem here.**

## D RESPONSE FILTER

We filter out generated responses by following rules.

- Delete the responses without an answer.
- Delete the responses that are excessively lengthy($> 1500$).
- Remove superfluous characters beyond the reasoning path and the answer.

## E DETAILED EXPERIMENTAL RESULTS

We list the detailed experimental results of different settings here.

| Query aug. type | 7B | 7B-2 | 13B-2 |
|---|---|---|---|
| No aug. ($\mathcal{D}$) | 35.9 | 41.6 | 50.0 |
| Change numbers | 41.5 | 48.5 | 54.1 |
| Fractions or percentages | 41.2 | 46.2 | 54.4 |
| Combine multiple concepts | 41.1 | 47.5 | 56.1 |
| Conditional statement | 41.7 | 45.8 | 56.4 |
| Increase complexity | 42.4 | 48.6 | 57.6 |
| All aug. ($\mathcal{D} + \mathcal{D}_1^1$) | **53.0** | **57.0** | **65.5** |

Table 6: Different query augmentation strategies on GSM8K performances.

| Query aug. | 7B | 7B-2 | 13B-2 |
|---|---|---|---|
| $\mathcal{D}$ | 35.9 | 41.6 | 50.0 |
| $+\mathcal{D}_1^1 \times 0.2$ | 38.1 | 43.6 | 56.3 |
| $+\mathcal{D}_1^1 \times 0.4$ | 45.0 | 51.3 | 58.9 |
| $+\mathcal{D}_1^1 \times 0.6$ | 48.2 | 55.8 | 61.9 |
| $+\mathcal{D}_1^1 \times 0.8$ | 51.1 | 56.6 | 63.2 |
| $+\mathcal{D}_1^1$ | 53.0 | 57.0 | 65.5 |
| $+\mathcal{D}_1^1 + \hat{\mathcal{D}}_2^1$ | 58.2 | 61.2 | 67.2 |
| $+\mathcal{D}_1^1 + \hat{\mathcal{D}}_2^1 + \mathcal{D}_3^1$ | **61.4** | **66.3** | **69.8** |

Table 7: The performance of SFT with different amounts of augmented query on GSM8K.

| Response aug. | 7B | 7B-2 | 13B-2 |
|---|---|---|---|
| $\mathcal{D}$ | 35.9 | 41.6 | 50.0 |
| $+\mathcal{D}_1^1$ | 53.0 | 57.0 | 65.5 |
| $+\mathcal{D}_1^1 + \mathcal{D}_1^2$ | 55.9 | 61.4 | 67.0 |
| $+\mathcal{D}_1^1 + \mathcal{D}_1^2 + \mathcal{D}_1^3$ | **61.3** | **64.4** | 68.4 |
| $+\mathcal{D}_1^1 + \mathcal{D}_1^2 + \mathcal{D}_1^3 + \mathcal{D}_1^4$ | 60.1 | 63.8 | 69.1 |
| $+\mathcal{D}_1^1 + \mathcal{D}_1^2 + \mathcal{D}_1^3 + \mathcal{D}_1^4 + \mathcal{D}_1^5$ | 60.7 | 63.6 | **71.6** |
| $+\mathcal{D}_1^1 + \mathcal{D}_1^2 + \mathcal{D}_1^3$ - majority voting | 56.4 | 60.7 | 65.7 |
| $+\mathcal{D}_1^1 + \mathcal{D}_1^2 + \mathcal{D}_1^3 + \mathcal{D}_1^4 + \mathcal{D}_1^5$ - majority voting | 58.9 | 62.5 | 68.3 |

Table 8: The performance of SFT with different amounts of augmented response on GSM8K.

## F CASE STUDY

## G DIFFICULTY LEVEL DEFINITION ON GSM8K

We conducted a statistical analysis of the reasoning paths required for 7,473 questions in the GSM8K training set, categorizing them as hard, medium, and easy. Specifically, we defined questions with more than three formulas, exactly three formulas, and less than three formulas as hard, medium, and easy, respectively. This categorization yielded a balanced distribution with 2,357 easy, 2,360 medium, and 2,756 hard problems. This approach ensures a relatively equal number of problems in each category.

| Model | Data | $\mathcal{D}$ | $+\mathcal{D}_1^1\times 0.2$ | $+\mathcal{D}_1^1\times 0.4$ | $+\mathcal{D}_1^1\times 0.6$ |
|---|---|---|---|---|---|
| 7B | training set | 56.4 | 41.8 | 51.4 | 55 |
| | test set | 35.9 | 38.1 | 45 | 48.2 |
| 7B-2 | training set | 65.2 | 48.4 | 57.4 | 64.8 |
| | test set | 41.6 | 43.6 | 51.3 | 55.8 |
| 13B-2 | training set | 75.4 | 80.4 | 80.4 | 82.6 |
| | test set | 50 | 56.3 | 58.9 | 61.9 |

(a) Part 1

| Model | Data | $+\mathcal{D}_1^1\times 0.8$ | $+\mathcal{D}_1^1$ | $+\mathcal{D}_1^1 + \hat{\mathcal{D}}_2^1$ | $+\mathcal{D}_1^1 + \hat{\mathcal{D}}_2^1 + \mathcal{D}_3^1$ |
|---|---|---|---|---|---|
| 7B | training set | 61.6 | 71.4 | 79.8 | 83.6 |
| | test set | 51.1 | 53 | 58.2 | 61.4 |
| 7B-2 | training set | 66.6 | 79 | 85.2 | 85.6 |
| | test set | 56.6 | 57 | 61.2 | 66.3 |
| 13B-2 | training set | 82.2 | 84.4 | 86.6 | 89.2 |
| | test set | 63.2 | 65.5 | 67.2 | 69.8 |

(b) Part 2

Table 9: The accuracy on the training dataset and test dataset for GSM8K.

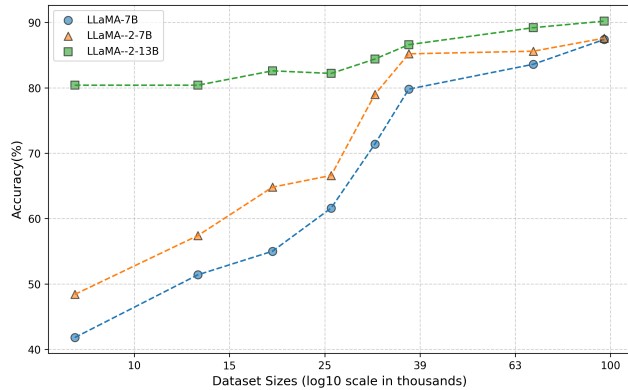

Figure 8: The relationship of accuracy on the training set and the amounts of augmentation data.

## H ADDED RESULTS

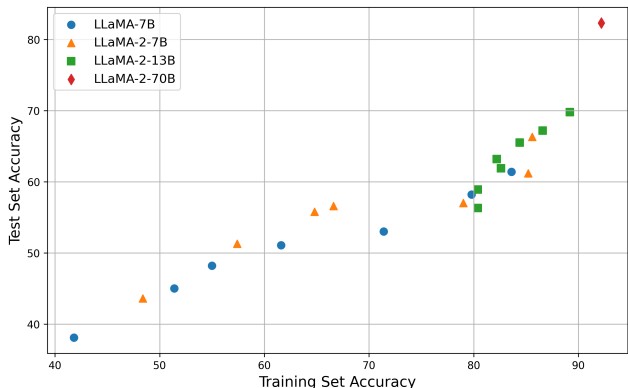

Figure 9: The relationship of accuracy on the GSM8K test set and the original GSM8K training set.

| Model | 7B | 7B-2 | 13B-2 |
|---|---|---|---|
| $\mathcal{D}$ | 35.9 | 41.6 | 50.0 |
| $\mathcal{D}_1^1$ on hard | 43.0 | 51.3 | 58.8 |
| $\mathcal{D}_1^1$ on medium | 43.5 | 49.0 | 55.6 |
| $\mathcal{D}_1^1$ on easy | 42.7 | 47.6 | 55.6 |
| $\mathcal{D}_1^1$ on random | 43.4 | 50.0 | 56.0 |

Table 10: The performance of SFT with query augmentation with different diffculties on GSM8K.

| Model | 7B | 7B-2 | 13B-2 |
|---|---|---|---|
| $\mathcal{D}_1^1$ on wrong | 46.2 | 49.5 | 55.4 |
| $\mathcal{D}_1^1$ on random | 43.6 | 49.2 | 54.2 |

Table 11: The performance of SFT with query augmentation with wrong problems or random problems on GSM8K.

| Method | Query |
|---|---|
| Original | Weng earns $12 an hour for babysitting. Yesterday, she just did 50 minutes of babysitting. How much did she earn? |
| Change specific numbers | Weng earns $15 an hour for babysitting. Last week, she babysat for 3 hours and 45 minutes every day. How much did she earn in total last week? |
| Introduce fractions or percentages | Weng earns $12 an hour for babysitting. However, for every hour she babysits beyond the first 3 hours, she earns an additional 20% per hour. If she babysat for 5 hours yesterday, how much did she earn? |
| Combine multiple concepts | Weng earns $12 an hour for babysitting, but gets paid only 75% of the total at the time of service. The rest is paid at the end of the month. If she babysat for 3 hours yesterday, how much was she paid immediately and how much will she receive later? |
| Include a conditional statement | Weng earns $12 an hour for babysitting. If it's a weekend, she gets a bonus of $5. If she babysat for 2 hours last Sunday, how much did she earn? |
| Increase the complexity of the problem | Weng earns $12 an hour for babysitting during weekdays and $15 an hour on weekends. If she babysat for 2 hours on Tuesday and 3 hours on Saturday, how much did she earn in total? |

Table 12: Examples of different methods for generating new queries.

| Query | Weng earns $12 an hour for regular babysitting but doubles her rate during weekends. Yesterday, she did 50 minutes of babysitting and it was a weekend. How much did she earn? |
|-------|-----------------------------------------------------------------------------------------------------------------------------------------------------------------------------------------|
| Response1 | Weng earns $12 per hour and doubles her rate during weekends, so she earned $12 * 2 = 24 per hour yesterday. There are 60 minutes in an hour, so Weng did 50 minutes / 60 minutes/hour = 0.833 hours of babysitting. Therefore, Weng earned $24 * 0.833 = 19.992, which we can round to $20. |
| Response2 | Weng earns $12*2 = 24 an hour during weekends. 50 minutes is 50/60 = 0.833 of an hour. So, Weng earned 24*0.833 = 19.992 dollars yesterday. |
| Response3 | During weekends, Weng earns 12*2 = 24 dollars per hour for babysitting. There are 60 minutes in an hour. So, 50 minutes is 50/60 = 0.833 of an hour. Weng earned 24*0.833 = 20 dollars for 50 minutes of babysitting during the weekend. |
| Response4 | Her weekend rate is 12*2 = 24 dollars per hour. There are 60 minutes in an hour, so she worked 50/60 = 0.833 hours. So, Weng earned 24*0.833 = 19.992 dollars. |
| Response5 | Weng's rate during weekends is 12*2 = 24 dollars per hour. Since an hour has 60 minutes, Weng earned 24*(50/60) = 20 dollars yesterday. |
| Response6 | Weng earns $12*2 = 24 an hour during weekends. She worked for 50/60 = 0.833 hour yesterday. So, Weng earned 0.833*$24 = 19.992 yesterday. |
| Response7 | Weng earns $12 an hour, but during the weekend, she doubles her rate which means she earns 12*2 = 24 dollars an hour during the weekend. Babysitting for 50 minutes is equivalent to 50/60 = 0.833 hours. Therefore, Weng earned 24*0.833 = 19.992 dollars from babysitting yesterday. |

Table 13: Different reasoning paths for the same query.

| Query | Response |
|---|---|
| Janet's ducks lay 16 eggs per day. She eats three for breakfast every morning and bakes muffins for her friends every day with four. She sells the remainder at the farmers' market daily for $2 per fresh duck egg. How much in dollars does she make every day at the farmers' market? | Janet eats 3 eggs for breakfast and bakes 4 eggs into muffins, so she uses 3+4 = 7 eggs every day. Janet's ducks lay 16 eggs per day, so after she uses 7 eggs, she has 16-7 = 9 eggs left. She sells these 9 eggs at the farmers' market for $2 per egg, so she makes 9*2 = 18 dollars every day at the farmers' market. |
| A robe takes 2 bolts of blue fiber and half that much white fiber. How many bolts in total does it take? | The white fiber is half the amount of the blue fiber so it takes 2/2 = 1 bolt of white fiber. All total it takes 2+1=3 bolts of fiber. |
| Josh decides to try flipping a house. He buys a house for $80,000 and then puts in $50,000 in repairs. This increased the value of the house by 150%. How much profit did he make? | He bought the house for 80,000 and put 50,000 in repairs so he had 80,000+50,000 = $130,000 invested. The value of the house increased by 150% so it went up 130,000*1.5 = $195,000 in value. He had 130,000 invested and it increased in value by 195,000 so he made a profit of 195,000-130,000 = $65,000. |
| James decides to run 3 sprints 3 times a week. He runs 60 meters each sprint. How many total meters does he run a week? | He runs 3*60=180 meters per day. So he runs 180*3=540 meters. |
| Every day, Wendi feeds each of her chickens three cups of mixed chicken feed, containing seeds, mealworms and vegetables to help keep them healthy. She gives the chickens their feed in three separate meals. In the morning, she gives her flock of chickens 15 cups of feed. In the afternoon, she gives her chickens another 25 cups of feed. How many cups of feed does she need to give her chickens in the final meal of the day if the size of Wendi's flock is 20 chickens? | Every day, Wendi feeds her chickens a total of 15+25=40 cups of feed. Thus, she must feed her chickens 40/3=13.333 cups of feed in the final meal of the day. |

Table 14: Some examples of MuggleMath-13B answering questions from the GSM8K test set. Wrong reasoning processes are labeled in red.

| Instruction | Response |
|---|---|
| A regular hexagon can be divided into six equilateral triangles. If the perimeter of one of the triangles is 21 inches, what is the perimeter, in inches, of the regular hexagon? | Since the triangle is equilateral, each side length is 7 inches. The hexagon has six sides of equal length, so its perimeter is $6 \times 7 = \boxed{42}$ inches. |
| What is the smallest positive perfect cube that can be written as the sum of three consecutive integers? | The response provided is incomplete and does not provide a valid answer to the question. |
| The set of points $(x, y, z)$ that satisfy $2x = 3y = -z$ is a line. The set of points $(x, y, z)$ that satisfy $6x = -y = -4z$ is another line. Find the angle between these lines, in degrees. | The direction vector of the first line is $< 2, 3, -1 >$, and the direction vector of the second line is $< 6, -1, -4 >$. The cosine of the angle between these direction vectors is $\frac{<2,3,-1>\cdot<6,-1,-4>}{\|<2,3,-1>\|\|<6,-1,-4>\|} = \frac{12}{20} = \frac{3}{5}$. Therefore, the angle between the lines is $\boxed{75°}$. |
| What is the distance, in units, between the points $(2, -6)$ and $(-4, 3)$? Express your answer in simplest radical form. | We use the distance formula: $\sqrt{(2-(-4))^2 + (-6-3)^2} = \sqrt{16 + 9^2} = \sqrt{16 + 81} = \sqrt{97} = \boxed{\color{red}3\sqrt{7}}$. |
| The expression $2 \cdot 3 \cdot 4 \cdot 5 + 1$ is equal to 121, since multiplication is carried out before addition. However, we can obtain values other than 121 for this expression if we are allowed to change it by inserting parentheses. For example, we can obtain 144 by writing $(2 \cdot (3 \cdot 4)) \cdot (5 + 1) = 144$. In total, how many values can be obtained from the expression $2 \cdot 3 \cdot 4 \cdot 5 + 1$ by inserting parentheses? (Note that rearranging terms is not allowed, only inserting parentheses). | We can obtain 121 by not inserting any parentheses, so we need to insert at least one pair of parentheses to obtain a different value. We can obtain 144 by inserting one pair of parentheses as shown above. If we insert two pairs of parentheses, we get $(2 \cdot (3 \cdot (4 \cdot 5))) + 1 = 121$. Thus, we can obtain 121 by inserting two pairs of parentheses, so we need to insert at least three pairs of parentheses to obtain a different value. We can obtain 168 by inserting three pairs of parentheses, since $(2 \cdot (3 \cdot (4 \cdot (5 + 1)))) = 168$. Thus, there are $\boxed{4}$ values that can be obtained from the expression $2\cdot3\cdot4\cdot5+1$ by inserting parentheses. |

Table 15: Some examples of MuggleMath-13B answering questions from the MATH test set. Wrong reasoning processes are labeled in red.

| closed-source models | | |
|---|---|---|
| Model | #params | GSM8K |
| GPT-4(OpenAI, 2023) | - | 92.0 |
| GPT-3.5-Turbo(Ouyang et al., 2022) | - | 80.8 |
| Claude-2 | - | 85.2 |
| PaLM (Chowdhery et al., 2022) | 8B | 4.1 |
| PaLM | 62B | 33.0 |
| PaLM | 540B | 56.5 |
| PaLM-2(Anil et al., 2023) | 540B | 80.7 |
| Flan-PaLM(Anil et al., 2023) 2 | 540B | 84.7 |
| Minerva(Lewkowycz et al., 2022b) | 8B | 16.2 |
| Minerva | 62B | 52.4 |
| Minerva | 540B | 58.8 |
| open-source models (1-10B) | | |
| LLaMA-1(Touvron et al., 2023a) | 7B | 11.0 |
| LLaMA-2(Touvron et al., 2023b) | 7B | 14.6 |
| MPT(Team, 2023b) | 7B | 6.8 |
| Falcon | 7B | 6.8 |
| InternLM(Team, 2023a) | 7B | 31.2 |
| GPT-J(Wang & Komatsuzaki, 2021) | 6B | 34.9 |
| ChatGLM-2(Zeng et al., 2022) | 6B | 32.4 |
| Qwen(Alibaba, 2023) | 7B | 51.6 |
| Baichuan-2(BaichuanInc, 2023) | 7B | 24.5 |
| SFT | 7B | 41.6 |
| RFTYuan et al. (2023a) | 7B | 50.3 |
| WizardMath(Luo et al., 2023a) | 7B | 54.9 |
| MetaMath(Yu et al., 2023) | 7B | 66.5 |
| **MuggleMath-7B** | 7B | 68.4 |
| **MuggleMath-new-7B** | 7B | **70.2** |
| open-source models (11-50B) | | |
| LLaMA-1(Touvron et al., 2023a) | 13B | 17.8 |
| LLaMA-1 | 33B | 35.6 |
| LLaMA-2(Touvron et al., 2023b) | 13B | 28.7 |
| LLaMA-2 | 34B | 42.2 |
| MPT(Team, 2023b) | 30B | 15.2 |
| Falcon(Penedo et al., 2023) | 40B | 19.6 |
| GAL(Taylor et al., 2023) | 30B | - |
| Vicuna(W. Chiang & Xing., 2023) | 13B | 27.6 |
| Baichuan-2(BaichuanInc, 2023) | 13B | 52.8 |
| SFT | 13B | 50.0 |
| RFTYuan et al. (2023a) | 13B | 54.8 |
| WizardMath(Luo et al., 2023a) | 13B | 63.9 |
| MetaMath(Yu et al., 2023) | 13B | 72.3 |
| **MuggleMath-13B** | 13B | **74.0** |
| **MuggleMath-new-13B** | 13B | **75.4** |
| open-source models (51-70B) | | |
| LLaMA-1(Touvron et al., 2023a) | 65B | 50.9 |
| LLaMA-2(Touvron et al., 2023b) | 70B | 56.8 |
| RFT(Yuan et al., 2023a) | 70B | 64.8 |
| WizardMathLuo et al. (2023a) | 70B | 81.6 |
| MetaMath(Yu et al., 2023) | 70B | 82.3 |
| **MuggleMath-70B** | 70B | **82.3** |

Table 16: Model comparison of MuggleMath and a broad range of state-of-the-art approaches. We conduct an extended experiments on larger datasets($\mathcal{D}$ +($\sum_{i=1}^{6} \mathcal{D}_1^i + \mathcal{D}_1^8$)-majority voting+$\mathcal{D}_2^1 + \mathcal{D}_3^1$) demonstrate to construct a new version of MuggleMath named MuggleMath-new, which achieves a better performance(70.2 for 7B size and 75.4 for 13B size on GSM8K).

|  | 7B | 7B-2 | 13B-2 |
|---|---|---|---|
| **Change-Test** | | | |
| SFT | 26.2 | 30.1 | 38.6 |
| MuggleMath | 60.1 | 62.8 | 67.1 |
| **Aug-Test** | | | |
| SFT | 14.2 | 17.2 | 22.4 |
| MuggleMath | 40.1 | 44.3 | 49.3 |

Table 17: We have perturbed two new test sets based on the original GSM8K test set. (A) Change-Test, is created by altering the numerical values in the GSM8K test set questions and correspondingly modifying the answers. There are 1211 query-response pairs in the Change-Test. (B) Aug-Test, is generated by augmenting the test set in the same manner as we did for the training set. There are 1378 query-response pairs in the Aug-Test.

| Subject | math | GSM8k | GSM8Kk+$\mathcal{D}_1^i$ | GSM8K+$\sum_{i=1}^{3}\mathcal{D}_1^i+\mathcal{D}_2^1+\mathcal{D}_3^1$ |
|---|---|---|---|---|
| Counting & Probability | 10.5 | 13.2 | 7.9 | 5.3 |
| Algebra | 7.3 | 12.1 | 12.9 | 16.9 |
| Prealgebra | 8.5 | 13.4 | 8.5 | 11.0 |
| Geometry | 2.4 | 9.8 | 4.9 | 2.4 |
| Intermediate Algebra | 6.2 | 5.2 | 3.1 | 5.2 |
| Number Theory | 3.2 | 6.5 | 6.5 | 8.1 |
| Precalculus | 3.6 | 5.4 | 7.1 | 7.1 |

Table 18: Transfer learning accuracy on subsets of MATH for LLaMA-13B-2

| Subject | math | GSM8k | GSM8Kk+$\mathcal{D}_1^i$ | GSM8K+$\sum_{i=1}^{3}\mathcal{D}_1^i+\mathcal{D}_2^1+\mathcal{D}_3^1$ |
|---|---|---|---|---|
| Prealgebra | 12.2 | 9.8 | 11.0 | 12.2 |
| Number Theory | 6.5 | 9.7 | 6.5 | 9.7 |
| Algebra | 7.3 | 5.6 | 5.6 | 15.3 |
| Intermediate Algebra | 2.1 | 4.1 | 4.1 | 1.0 |
| Precalculus | 3.6 | 1.8 | 3.6 | 1.8 |
| Counting & Probability | 5.3 | 7.9 | 5.3 | 13.2 |
| Geometry | - | 2.4 | - | - |

Table 19: Transfer learning accuracy on subsets of MATH for LLaMA-7B-2

| Subject | math | GSM8k | GSM8Kk+$\mathcal{D}_1^i$ | GSM8K+$\sum_{i=1}^{3}\mathcal{D}_1^i+\mathcal{D}_2^1+\mathcal{D}_3^1$ |
|---|---|---|---|---|
| Prealgebra | 7.3 | 9.8 | 7.3 | 14.6 |
| Number Theory | 6.5 | 3.2 | 1.6 | 3.2 |
| Algebra | 6.5 | 4.8 | 11.3 | 4.8 |
| Intermediate Algebra | 3.1 | 1.0 | 5.2 | 3.1 |
| Precalculus | - | 3.6 | 5.4 | 1.8 |
| Counting & Probability | 2.6 | 2.6 | 2.6 | 7.9 |
| Geometry | 4.9 | 4.9 | 2.4 | 2.4 |

Table 20: Transfer learning accuracy on subsets of MATH for LLaMA-7B

| Subject | math | GSM8k | GSM8Kk+$\mathcal{D}_1^1$ |
|---|---|---|---|
| Prealgebra | 8.5 | 12.2 | 14.6 |
| Number Theory | 3.2 | 12.9 | 3.2 |
| Algebra | 7.3 | 8.1 | 12.1 |
| Intermediate Algebra | 6.2 | 5.2 | 7.2 |
| Precalculus | 3.6 | - | - |
| Counting & Probability | 10.5 | 5.3 | 10.5 |
| Geometry | 2.4 | 7.3 | 4.9 |

Table 21: Multi-task learning accuracy on subsets of MATH for LLaMA-13B-2

| Subject | math | GSM8k | GSM8Kk+$\mathcal{D}_1^1$ |
|---|---|---|---|
| Prealgebra | 12.2 | 11.0 | 4.9 |
| Number Theory | 6.5 | 6.5 | 3.2 |
| Algebra | 7.3 | 5.6 | 11.3 |
| Intermediate Algebra | 2.1 | 6.2 | - |
| Precalculus | 3.6 | - | - |
| Counting & Probability | 5.3 | 5.3 | 2.6 |
| Geometry | - | 7.3 | 7.3 |

Table 22: Multi-task learning accuracy on subsets of MATH for LLaMA-7B-2

| Subject | math | GSM8k | GSM8Kk+$\mathcal{D}_1^1$ |
|---|---|---|---|
| Prealgebra | 7.3 | 6.1 | 7.3 |
| Number Theory | 6.5 | 3.2 | 3.2 |
| Algebra | 6.5 | 6.5 | 10.5 |
| Intermediate Algebra | 3.1 | 2.1 | - |
| Precalculus | - | 1.8 | - |
| Counting & Probability | 2.6 | 2.6 | 5.3 |
| Geometry | 4.9 | 9.8 | 2.4 |

Table 23: Multi-task learning accuracy on subsets of MATH for LLaMA-7B

