# OpenReview forum: "Query and Response Augmentation Cannot Help Out-of-domain Math Reasoning Generalization"
_ICLR.cc/2024/Conference — ICLR 2024 Conference Withdrawn Submission_

### Official Review · Reviewer_q1A4 · 2023-10-19

**Soundness:** 3 good
**Presentation:** 2 fair
**Contribution:** 3 good
**Rating:** 6
**Confidence:** 4

**Summary:**

The paper presented data augmentation method for LLM and studied augmentation is log linear to the amount of data added in the training. It found that augmentation didn't help on OOD case. Overall very interesting study and provided insights on data augmentation method.

**Strengths:**

- The experiments are comprehensive. The authors conducted lots of experiments to study data augmentation impact.
- The analysis is well-organized and reasoning is sound

**Weaknesses:**

- Writing can be improved. Figure 1 caption has duplicated `LLaMA-2-7B`.

**Questions:**

In the paper, it shows that even on wrong problem, data augmentation still helps. It would be interesting to know if we have some perturbation on the test, will there be still any improvement?

---

> ### Author Response · Authors · 2023-11-15
> **We have created two new perturbed test sets based on GSM8K and report that their MuggleMath model outperforms the one fine-tuned only on GSM8K.**
>
> Thanks for your insightful comment.
>
> W1: We have fixed the typo in the Figure 1 caption.
>
> W2: We have perturbed two new test sets based on the original GSM8K test set.
>
> (A) Change-Test, is created by altering the numerical values in the GSM8K test set questions and correspondingly modifying the answers. There are 1211 query-response pairs in the Change-Test.
>
> (B) Aug-Test, is generated by augmenting the test set in the same manner as we did for the training set. There are 1378 query-response pairs in the Aug-Test.
>
> Upon evaluating our model on these two perturbed test sets, we found that the performance of MuggleMath consistently and significantly exceeds that of the model fine-tuned on GSM8K alone. This observation suggests that our data augmentation techniques not only enhance the model’s ability to solve the original problems but also contribute to its improved performance on varied and perturbed inputs, thereby indicating a robust generalization capability. The results are added in in Appendix H.
>
> |               | **7B** | **7B-2** | **13B-2** |
> |---------------|:------:|:--------:|:---------:|
> |               |        |          |           |
> | **Change-Test**       |        |          |           |
> | SFT           | 26.2   | 30.1     | 38.6      |
> | MuggleMath    | 60.1   | 62.8     | 67.1      |
> |               |        |          |           |
> | **Aug-Test**      |        |          |           |
> | SFT           | 14.2   | 17.2     | 22.4      |
> | MuggleMath    | 40.1   | 44.3     | 49.3      |

---

### Official Review · Reviewer_MHWk · 2023-11-06

**Soundness:** 3 good
**Presentation:** 2 fair
**Contribution:** 3 good
**Rating:** 5
**Confidence:** 3

**Summary:**

This paper presents a study on data augmentation in math reasoning, investigating the effectiveness of different strategies and the scaling relationship between the amount of augmented data and model performance. The authors created a new dataset, AugGSM8K, and obtained a series of LLMs called MuggleMath that achieved new state-of-the-art on GSM8K.

**Strengths:**

The authors conducted an investigation of different data augmentation strategies for math reasoning, including query augmentation, response augmentation, and both query and response augmentation. They evaluated the effectiveness of these strategies on the AugGSM8K dataset and analyzed the scaling relationship between the amount of augmented data and model performance.

The authors achieved new state-of-the-art results on the GSM8K dataset with their LLMs called MuggleMath. They fine-tuned MuggleMath on subsets of the AugGSM8K dataset. The authors' investigation of data augmentation strategies and their analysis of the scaling relationship between the amount of augmented data and model performance can help inform future research in this area and contribute to the development of more robust and generalizable models for math reasoning.

**Weaknesses:**

The authors did not compare their approach with other state-of-the-art approaches for math reasoning, which makes it difficult to assess the relative effectiveness of their approach. The authors generated multiple reasoning paths for each augmented problem, but did not provide a detailed analysis of the impact of different reasoning paths on model performance. A more in-depth analysis of reasoning paths could help identify which types of reasoning paths are most effective for different types of problems.

The impact of data augmentation on model interpretability was not thoroughly analyzed in the paper. The effects of data augmentation on the models' ability to provide explanations or justifications for their reasoning remain unclear.

Furthermore, the authors did not conduct a comprehensive review of the computational efficiency of their approach. Although they mentioned using proprietary models (GPT-3.5 and GPT-4) to implement five types of mathematical problem augmentation methods based on human experience in creating variations of mathematical problems, these models are known to be computationally intensive and demand significant computational resources. The computational cost of their approach could potentially limit its scalability to larger datasets or practical applications. A more thorough analysis of the computational efficiency of their approach could help identify potential bottlenecks and inform the development of more efficient models for mathematical reasoning.

**Questions:**

The authors did not juxtapose their approach with other cutting-edge methods for mathematical reasoning, making it difficult to gauge the relative efficacy of their solution. They generated numerous reasoning paths for each augmented problem, yet they did not provide an in-depth analysis of how different reasoning paths affect model performance. A deeper exploration of these reasoning paths could potentially determine the most effective paths for various problem types.

The influence of data augmentation on model interpretability was not adequately scrutinized in the paper. The implications of data augmentation on the capacity of models to furnish explanations or justifications for their reasoning are still ambiguous.

Moreover, the authors did not undertake a comprehensive assessment of the computational efficiency of their methodology. They mentioned the use of proprietary models (GPT-3.5 and GPT-4) to apply five types of mathematical problem augmentation methods, reflecting human experience in creating problem variations. These models are renowned for their computational intensity and substantial resource requirements. The computational expense of their methodology could potentially restrict its scalability to larger datasets or practical applications. A more exhaustive analysis of the computational efficiency of their approach could aid in identifying potential bottlenecks and guide the creation of more efficient models for mathematical reasoning.

---

> ### Author Response · Authors · 2023-11-15
>
> Thanks for your insightful comment.
>
> __W1: We have included additional comparisons with a broader range of state-of-the-art approaches in Appendix H.  Part of comparisons are list here.__
>
> Further, We conduct an extended experiments on larger datasets($\mathcal{D}$ +($\sum_{i=1}^{6}{\mathcal{D}_1^i}+\mathcal{D}_1^8$)-majority voting+$\mathcal{D}_2^1+\mathcal{D}_3^1$) demonstrate to construct a new version of MuggleMath named __MuggleMath-new__, which achieves a better performance(__70.2 for 7B size and 75.4 for 13B size on GSM8K__).
>
> | Model                       | #params | GSM8K |
> |-----------------------------|---------|-------|
> | closed-source models    |         |       |
> | GPT-4                       | -       | 92.0  |
> | GPT-3.5-Turbo               | -       | 80.8  |
> |Claude-2                        | - | 85.2 |
> | open-source models (1-10B) |     |       |
> | LLaMA-2                     | 7B      | 14.6  |
> | MetaMath                    | 7B      | 66.5  |
> | **MuggleMath-7B**           | 7B      | **68.4**  |
> | **MuggleMath-new-7B**       | 7B      | **70.2** |
> | open-source models (11-50B) |    |       |
> | LLaMA-2                     | 13B     | 28.7  |
> | LLaMA-2                     | 34B     | 42.2  |
> | MetaMath                    | 13B     | 72.3  |
> | **MuggleMath-13B**          | 13B     | **74.0** |
> | **MuggleMath-new-13B**      | 13B     | **75.4** |
> | open-source models (51-70B)|    |       |
> | LLaMA-2                     | 70B     | 56.8  |
> | MetaMath                    | 70B     | 82.3  |
> | **MuggleMath-70B**          | 70B     | **82.3** |
>
>
>
>
> __W2: Regarding the impact of reasoning paths on performance, Section 3.3 of our paper examines the response effectiveness for the same problems.__
>
> We conclude that, compared to the zero-shots generation method, the response augmentation prompt we use plays a substantial role(+3.6 for LLaMA-7B, +3.7 for LLaMA-2-7B, +3.3 for LLaMA-2-13B), since the 1-shot setting stabilizes the response format.
> We find that, compared to GPT-3.5, the responses augmented using GPT-4 perform significantly better for SFT. Hence, the most effective way to generate the response is to use the best model and well-designed prompts.
>
> __W3: In response to the concern about the impact of data augmentation on model interpretability.__
>
> Our augmentation process involves creating variations of the original problems, which, as shown in Figure 1, results in the augmented queries from AugGSM8K having a similar distribution in the model’s latent space as the original GSM8K queries. Generally, in deep learning, more data from the same distribution tends to improve model performance, which is a key reason why models trained on AugGSM8K exhibit superior performance on the original GSM8K dataset.
>
> In addition, we have conducted an analysis of our model’s performance on test set problems of varying difficulty. Our analysis of the 7B-2 model’s performance on the test set, shows accuracy rates of 0.55, 0.42, and 0.21 for easy, medium, and hard problems, respectively. The MuggleMath model, however, achieved higher accuracy rates of 0.73, 0.70, and 0.64 for the same problem categories.This significant performance boost on difficult questions can be attributed to the fact that the augmented problems we generated are generally more complex than the original problems.
>  | Model      | Easy  | Medium | Hard  |
> |------------|-------|--------|-------|
> | 7B-2   SFT    | 0.55  | 0.42   | 0.21  |
> | MuggleMath-7B-2 | 0.73  | 0.70   | 0.64  |
> Moreover, as discussed in Section 4.2 and illustrated in Figure 2, training the model on more challenging problems (those requiring more reasoning steps) and on problems it initially solved incorrectly leads to a more efficient improvement in the model’s mathematical abilities. This observation mirrors the way humans tend to benefit from practicing more difficult math problems and learning from their mistakes.
>
> __W4: In response to the concern about  computational efficiency of our methodology.__
>
> During the data generation phase, which includes the generation of queries and responses, the cost associated with using the GPT-3.5-turbo API is 0.002 dollars per 1k tokens, and for the GPT-4 API, the price is 0.03 dollars per 1k input tokens and 0.06 dollars per 1k output tokens. As shown in Table 1, the creation of the 10 versions of AugGSM8K resulted in a total expenditure of 4057.2 dollars. Specifically, in Table 4 for MuggleMath in Table 4, the cost for data generation amounted to 2508.9 dollars.
>
> In the training phase, to train the models that achieved the results presented in Table 4 for MuggleMath, the computational resources utilized were as follows: the 7B, 7B-2, and 13B-2 models were trained on 8 NVIDIA A100 GPUs for 5.5 hours, 5.5 hours, and 13.5 hours, respectively. For the 70B-2 model, we trained on 32 NVIDIA A100 GPUs for 20 hours, which will be updated in the paper.

---

### Official Review · Reviewer_12bj · 2023-11-19

**Soundness:** 3 good
**Presentation:** 2 fair
**Contribution:** 1 poor
**Rating:** 3
**Confidence:** 3

**Summary:**

The paper studies data augmentation for mathematical reasoning by LLMs. In particular, the authors derive several augmentations for one of the standard benchmarks, GSM8k, by prompting GPT 3.5 and GPT 4 to generate more diverse queries and responses. It is then shown that fine-tuning on that augmented version of GSM8k brings improvements for the original GSM8k but does not really work for MATH, another benchmarking dataset for math reasoning.

**Strengths:**

**S1.** The authors created an augmented version of GSM8k that might be of some value for the benchmarking community.

**Weaknesses:**

**W1.** The paper lacks novelty and scientific contributions. The main message of the paper can be condensed to “Data augmentation works for IID setups but does not work in OOD setups” - this is well-known in the ML theory and I do not see what new contributions this paper brings to this problem. GSM8k and MATH are different enough to assume by default that augmentations of one dataset are unlikely to bring benefits for another.

**W2.** The paper mostly cites LLM papers from 2021-2023 as if years of works on the ML theory of augmentation and their generalization capabilities do not exist. This is quite narrow-minded and leads to reinventing the wheel but now under the LLM umbrella. I would expect a more rigorous study of why augmentations do not help on MATH (and other math reasoning datasets if available) backed by the theory instead of hand-wavy conclusions “apply augmentations on all datasets” and “improve pre-traininig” offered by the authors.

**W3.** What is the value of the “scaling laws” for the augmented GSM8k (where overall sizes are 4K-100K data points)? The authors mention that the derived log-linear lines are unlikely to transfer to other datasets, so the lines merely show that bigger LLMs bring better performance, but there is no deeper implication of that.

**Questions:**

Q1. What are the "hidden representations of problems encoded by LLaMA2-7B" used for t-SNE given the causal architecture of Llama?

---

> ### Author Response · Authors · 2023-11-21
>
> We appreciate the reviewer's review.
>
> For W1, there is a trend in LLM community augmenting many more samples to race on math benchmark performance. As for the novelty and scientific contributions of our paper, we would like to highlight that our work builds on the advantages of previous works such as RFT (response augmentation) and WizardMath (query augmentation), which have been widely recognized and cited in the LLM community. Our study not only achieved the best performance among all open-source models on the GSM8K dataset but also provided an analysis of scaling laws and out-of-distribution generalization. This analysis can offer valuable insights to the community when using data augmentation methods to enhance models' mathematical reasoning abilities.
>
> For W2, we have taken your advice and add more ML generalization paper to related works.  Moreover, we have changed the conclusion to "apply augmentations on diverse math subjects".
>
> For W3, we consider scaling laws during fine-tuning augmented dataset is useful for deliver a specific ability (like math reasoning or coding) to users. And another finding here is augmenting using model-generated samples and using human-written samples have similar scaling trend which is very interesting for who want to improve the model based on model itself.
>
> For Q1, we use the 15-th layer of last token representation in the problem.

---

### Official Review · Reviewer_gKX2 · 2023-11-20

**Soundness:** 3 good
**Presentation:** 2 fair
**Contribution:** 1 poor
**Rating:** 5
**Confidence:** 3

**Summary:**

In this article the authors investigate the influence of a data augmentation technique on the performance of large language models for mathematical reasoning tasks. They augment the GSM8K dataset by prompting proprietary LLMs to 1) create variations of queries, changing one of 5 aspects of the query and 2) create variations of the responses. The authors derive scaling laws for model performance based on the amount of augmented data. They find a linear relationship between test set and training set accuracy and furthermore investigate the effect of multitask training with augmented data on performance on the MATH dataset, finding no evidence for improvement due to data augmentation on GSM8K.

**Strengths:**

-	The authors created AugGSM8K a LLM based augmentation of queries, that vary the original queries along 5 aspects.
-	The article contains a wide range of experiments, and the trained models achieve state of the art performance among open-source approaches.
-	The authors replicated (from earlier research) log-linear scaling laws of model performance on amounts of augmented queries and linear relationship between test set accuracy and training set accuracy for the chosen augmentation method.

**Weaknesses:**

-	The contributions of the paper are limited in the light of recent research. Similar scaling laws have been shown for similar augmentation techniques for the same datasets. The analysis of answer augmentation and query augmentation as well as their combination has also been explored in  Yu et al. 2023 "MetaMath: Bootstrap Your Own Mathematical Questions for Large Language Models" which has been cited in this article.
-	The evaluation on Out-of-Domain Generalization lacks evidence to support the strong claim in the title of the paper. Furthermore, the authors seem to be agnostic of previous research on Out-of-Domain generalization its well known challenges, and existing approaches. The title is also in so far misleading as it suggests there is no generalization effect between domains of mathematics (e.g., algebra vs calculus).
-	It is not clear how the reasoning paths have been sampled. A Case study on the prompting methods would improve the exposition.
- The writing of the paper could be improved.

**Questions:**

-	According to Fig. 5 there is an overlap between the distribution of MATH and GSM8K in the embedding space. Does the proposed augmentation method improve performance on this subset of MATH?
-	What is the generalization to other subsets of MATH (as listed in fig. 5)?
-	How do you employ human expertise and knowledge in mathematical problems for query augmentation (see paragraph 3.2)?

---

> ### Author Response · Authors · 2023-11-22
>
> We appreciate the reviewer's review.
>
> For W1, we acknowledge the work of Yu et al. 2023 “MetaMath: Bootstrap Your Own Mathematical Questions for Large Language Models”, which was indeed contemporaneous with our own research. Their paper was uploaded to arxiv on September 21, 2023, merely a day before our submission to ICLR. Consequently, our work was conducted independently and without the influence of Yu et al.'s findings. Our paper shares similar contributions with that of Yu et al. in terms of query and response enhancement methods and the analysis of scaling relationships. However, we have conducted further analysis on the generalizability of data augmentation techniques.
>
> For W2, our paper’s title, “Query and Response Augmentation Cannot Help Out-of-Domain Math Reasoning Generalization,” is primarily based on our findings that applying query and response augmentation to the GSM8K dataset resulted in negligible and inconsistent performance improvements on the MATH dataset.  There has been substantial research on Out-of-Domain (OOD) issues, and we will include a discussion of this literature in the related work section of our manuscript. Since data augmentation has demonstrated exceptional performance in large model math reasoning, but its generalizability has not been studied before, our paper provides a preliminary analysis of the generalizability of these methods. We believe this will offer some new insights to the community. To further investigate how the data augmentation performed on GSM8K affects different domains of mathematics, we conducted an additional analysis on the accuracy across various subsets of the MATH dataset.
>
> For W3, the reasoning paths are sampled by GPT3.5 or GPT4, which is discussed in section 3.2 and section 4.1. The prompt used for reasoning path generation is listed in appendix C and a case study can be seen in Table 13.
>
> For Q1 and Q2, while there is an overlap in the embedding space distribution between MATH and GSM8K, it is relatively small compared with that between GSM8K and AugGSM8K. In the transfer learning setting, training first on GSM8K and then on MATH with LLaMA-13B-2 does provide some benefits for certain subsets, such as Prealgebra, Algebra, and Geometry. However, if we train first on AugGSM8K and then on MATH, this benefit is not only marginal but may even lead to a decrease in performance on other subsets, like Geometry and Prealgebra, which could be related to the data proportions. Overall, the performance improvement on the MATH dataset from augmentation on GSM8K is minimal, even on subsets like Prealgebra, where there is some overlap. For 7B size and multi-task learning setting, we can draw the similar conclusion. Complete comparison will be listed in appendix H.
> | Subject               | math | GSM8K | +${\mathcal{D}_1^1}$ |+$\sum_{i=1}^{3}{\mathcal{D}_1^i}+\mathcal{D}_2^1 +\mathcal{D}_3^1$  |
> |-----------------------|------|-------|------------|------------|
> | Counting & Probability | 10.5 |  13.2 |     7.9    |     5.3    |
> | Algebra               | 7.3  |  12.1 |    12.9    |    16.9    |
> | Prealgebra            | 8.5  |  13.4 |     8.5    |    11.0    |
> | Geometry              | 2.4  |   9.8 |     4.9    |     2.4    |
> | Intermediate Algebra  | 6.2  |   5.2 |     3.1    |     5.2    |
> | Number Theory         | 3.2  |   6.5 |     6.5    |     8.1    |
> | Precalculus           | 3.6  |   5.4 |     7.1    |     7.1    |
>
>
> For Q3, in our approach to query augmentation, we took inspiration from the common pedagogical practice where human learners engage in “varied practice” training to enhance their mathematical problem-solving skills. Specifically, we employed several variation training methods commonly used in algebra to construct new problems based on the original ones.

---

> > ### Comment · Reviewer_gKX2 · 2023-11-26
> >
> > I appreciate the authors comments and I also acknowledge the comments to the other reviews.
> >
> > regarding W1 I understand that research in this field is dynamic. However, I think the overlap between the mentioned publication is to significant leading to little novelty of the article under considertion. Furthermore, the analysis of generalizability of data augmentation techniques is to shallow to constitue a sufficient contribution for publication at ICLR.
> >
> > I thank the authors for answering and clarifying my questions.
> >
> > Considering the overlap with "MetaMath:..." my initial assessment remains unchanged.

---

> ### Author Response · Authors · 2023-11-27
> **Replying to Official Comment by Reviewer gkX2**
>
> We appreciate the reviewer’s comments. We respectfully disagree with the assessment regarding the novelty of our work and feel that the perceived overlap with “MetaMath” does not fairly represent our contributions, for the following reasons:
>
> 1. **Our research was conducted in parallel with “MetaMath ...”**,  with our submission to ICLR taking place just one day after “MetaMath” was uploaded to arXiv, ensuring our work’s independence.
>
> 2. **Our data augmentation methods in MuggleMath differ from MetaMath in nature and have demonstrated superior performance on GSM8K** (MuggleMath-new-7B scored 70.2 vs. MetaMath-7B’s 66.5, and MuggleMath-new-13B scored 75.4 vs. MetaMath-13B’s 72.3). While MetaMath focuses on rewriting the question from multiple perspectives, our techniques not only alter numerical relationships but also introduce new logical content, enhancing the complexity and variety of problems.
>
> 3. Though initial, our analysis on the generalizability of data augmentation techniques **provides some new insights to the community**.

---

### Author Response · Authors · 2023-11-23
**Request for Reviewing Our Response and Revised Paper**

Dear reviewers, senior area chairs area chairs  and program chairs,

We kindly request that you take a moment to review our revised paper and assess whether our response addresses your concerns. If you have any additional questions or require further clarification, please do not hesitate to inform us. We value the opportunity to continue the discussion if necessary.

Best regards,

Submission 4983 Authors